# To the Best of Trust: Full-Stage Trusted Multi-modal Clustering

## Abstract

Multi-modal clustering (MMC) aims to integrate complementary information from different modalities to uncover latent consistent structures and improve clustering performance.However, existing methods mainly rely on predictive (result) uncertainty to improve robustness, while often neglecting aleatoric (data) uncertainty introduced by sample noise and epistemic (model) uncertainty induced by model parameters and structural variations.To this end, we propose a novel Full-Stage Trusted Multi-modal Clustering (FSTMC) method. To the best of trust, we jointly utilize aleatoric, epistemic, and predictive uncertainties to optimize the model, learn more reliable feature representations, and obtain more reliable clustering results. In the representation learning stage, probabilistic modeling is employed to capture stable latent representations that account for aleatoric uncertainty, while structured stochastic perturbations are introduced to estimate epistemic uncertainty. In the clustering stage, we replace conventional feature-level fusion with an evidence-based strategy: soft labels from each modality are mapped into categorical evidence, class distributions are parameterized via a Dirichlet model, and dynamic cross-modal fusion is achieved through Dempster–Shafer theory. To mitigate overconfidence and modal conflicts, prior constraints guided by aleatoric and epistemic uncertainty are imposed, resulting in calibrated predictive uncertainty. Finally, we exploit predictive uncertainty to selectively incorporate pseudo labels for optimization. Benchmark experiments on a large number of multi-modal datasets demonstrate that our approach significantly improves credibility and accuracy compared to state-of-the-art methods.

## 1 Itroduction

Multi-modal clustering (MMC) aims to extract complex nonlinear latent representations within modalities and explore potential connections between different modalities. By integrating complementary information from different perspectives, MMC can reveal hidden features that are difficult to detect in a single modality. This capability makes it show great potential in practical scenarios such as cross-modal retrieval (Zeng et al., 2021), sentiment analysis (Tang et al., 2024), and drone monitoring (Cao et al., 2024). In recent years, the rapid development of deep learning has significantly improved the representational capabilities and clustering effects of MMC. For example, some studies (Chen et al., 2021; Hu et al., 2023; Lou et al., 2025a) employ contrastive learning to enhance intra-modal feature representations and capture inter-modal consistency, while others (Yang et al., 2023b; Zou et al., 2023; Zhao et al., 2025) adopt graph neural networks to propagate information between nodes and edges, thereby modeling more complex and high-order cross-modal relationships.

However, existing methods generally overlook modality quality and the credibility of clustering results. To address this limitation, reliable MMC based on Dempster–Shafer (DS) evidence theory (Gordon & Shortliffe, 1984) has emerged, aiming to enhance interpretability and robustness by dynamically evaluating modality quality and quantifying predictive uncertainty. Xu et al. (2024) were the first to introduce trustworthy learning into MMC to handle conflicting instances with credibility estimation; subsequently, Cheng et al. (2024) leveraged the information bottleneck principle to extract reliable evidence and achieve unified modeling of beliefs and uncertainties. Building on this, Hu et al. (2025b) further refined uncertainty in self-supervised contrastive clustering by dynamically filtering pseudo-labels, thereby yielding more robust feature comparisons.

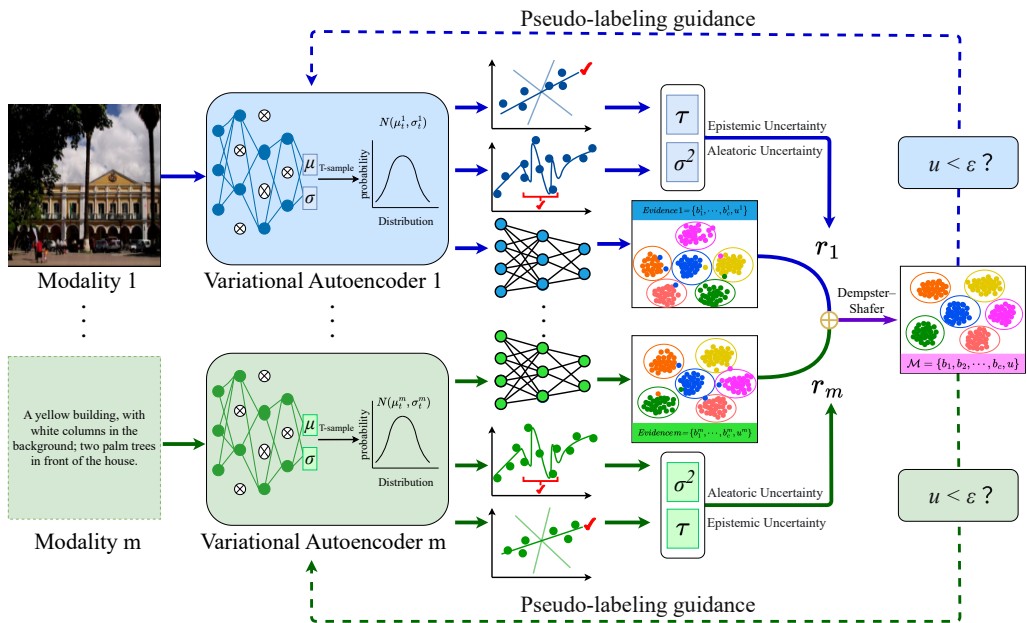

Figure 1: The framework of FSTMC. We first employ a variational autoencoder to map multimodal data into a latent distribution $\{\mathcal{N}(z; \mu, (\sigma)^2)\}_i^m$. We explicitly model two types of uncertainty: stochastic uncertainty $\{(\sigma)^2\}_i^m$ caused by data noise and epistemic uncertainty $\{\tau\}_i^m$ caused by model parameters. The latent features are then fed into the evidence module to generate a quality function $\{W\}_i^m$, which consists of a class belief quality $\{b_k\}_i^m$ and a predictive uncertainty $\{u\}_i^m$. To achieve more robust multimodal fusion, we construct an uncertainty-guided prior gating mechanism $\{r\}_i^m$ and leverage the Dempster-Shafer theory to aggregate evidence from different modalities. Ultimately, we obtain a soft cluster assignment, with the highest confidence class as the final output. In addition, we introduce a pseudo-labeling strategy, where only when the uncertainty is lower than a predefined threshold $\epsilon$, the pseudo-label is considered credible and used for further network optimization.

Although existing MMC methods have made some progress, they still have the following serious shortcomings: 1) These methods do not explicitly model or exploit data uncertainty arising from noise or sample quality and model uncertainty caused by parameter fluctuations and structural perturbations, making it difficult to maintain reliability during training. This often leads to disturbed gradients, delayed convergence, and ultimately reduced stability and robustness of the results. 2) These methods also fail to incorporate uncertainty modeling throughout the entire process of representation learning and clustering optimization. By neglecting instance-level quality differences, they underutilize high-quality features while allowing low-quality features to persistently interfere, which in turn causes overconfidence and cross-modal conflicts.

To address these limitations, we propose a novel Full-Stage Trusted Multi-modal Clustering (FSTMC) method, as shown in Figure 1. The core idea is to jointly model and exploit aleatoric, epistemic, and predictive uncertainties throughout the clustering process, thereby learning more robust feature representations and achieving more trustworthy clustering results. During representation learning, we employ probabilistic modeling to capture the low-dimensional latent structure of the data while characterizing aleatoric uncertainty from noise and sample quality, and introduce stochastic priors over model parameters to estimate epistemic uncertainty from parameter perturbations. This joint modeling improves representation reliability and provides explicit uncertainty quantification that benefits downstream clustering. In the clustering stage, we adopt an evidence fusion paradigm: soft labels from each modality-specific clustering network are mapped into class-wise evidence, parameterized as a Dirichlet distribution, and aggregated using Dempster–Shafer theory, which mitigates inter-modal conflicts and enhances robustness. Furthermore, aleatoric and epistemic uncertainties guide prior constraints to adaptively adjust modality weights, amplifying the contribution of reliable modalities while suppressing noisy ones. Finally, predictive uncertainty is

leveraged in a feedback loop to filter pseudo-labels, ensuring that only reliable supervision signals are used, thus realizing full-stage trustworthiness.

Our key contributions are as follows:

- We propose a novel FSTMC framework, which for the first time enables the joint quantification and exploitation of stochastic, epistemic, and predictive uncertainties in MMC, thus ensuring reliable clustering at all stages.

- We independently estimate the uncertainty within each modality, reduce the interference of noise and parameter uncertainty on feature representation, and at the same time reduce cross-modal conflicts during the fusion stage to improve overall robustness.

- We introduce aleatoric and epistemic uncertainty uncertainty in the evidence fusion stage to dynamically adjust modal weights and leverage prediction uncertainty to filter pseudo-labels, achieving dynamic closed-loop optimization in the feature learning and clustering stages.

- We conduct large-scale experimental validation on multiple multi-modal benchmark datasets, demonstrating that FSTMC significantly outperforms existing MMC methods in both performance and reliability.

## 2 THREE TYPES OF UNCERTAINTY IN FSTMC

**Aleatoric uncertainty:** Caused by noise and quality defects in the data itself, it belongs to the inherent randomness at the observation level and is difficult to eliminate directly by simply increasing the number of training samples. Intuitively, it limits the upper confidence limit that a single sample can reach: the stronger the noise, the more incomplete the information, the more ambiguous the discrimination space, and the wider the confidence interval.

**Epistemic uncertainty:** Originates from the "reducible" uncertainty of the model and knowledge itself, commonly seen in situations such as insufficient data coverage, inadequate parameter learning, model misspecification, or distribution shift. It manifests as the sensitivity of the latent representation or decision to reasonable perturbations, and is usually higher in low-density regions and between-class boundaries. Unlike aleatoric uncertainty, it can be reduced by supplementing with effective data, improving the structure and priors, and sufficient training and calibration.

**Predictive uncertainty:** The dispersion/dispersion of the final predicted posterior for a specific sample, characterizing "whether this prediction is reliable". In a multimodal context, it comprehensively reflects the strength and consistency of evidence from different sources: when evidence is concentrated and consistent in direction, the posterior is sharper and the confidence is higher; when evidence is sparse or contradictory, the posterior is flatter or the conflict is more significant, and the confidence decreases accordingly.

In our model, aleatoric, epistemic, and predictive uncertainty are interconnected and indispensable. Aleatoric reflects the upper limit of discriminability at the observation end, Epistemic reflects the room for improvement at the model end, and Predictive is the expression of uncertainty in a specific decision. In practice, increased observational noise is often accompanied by more unstable representations, locally showing a positive correlation between Aleatoric and Epistemic; an increase in either will increase predictive uncertainty of the current prediction. Meanwhile, if cross-modal evidence is consistent and of high quality, Predictive usually decreases; if evidence is sparse or significantly divergent, Predictive usually increases.

## 3 THE PROPOSED METHOD

### 3.1 PROBLEM FORMULATION

Let $\mathcal{X} = \{X^1, \ldots, X^M\}$ denote the datasets from $M$ modalities. For the $m$-th modality, we write $X^m = \{x_1^m, \ldots, x_n^m\} \in \mathbb{R}^{n \times d^m}$, where $n$ is the number of samples and $d^m$ is the feature dimension. Our method aims to comprehensively leverage aleatoric, epistemic, and predictive uncertainty to accurately assign samples to $K$ clusters while ensuring reliability throughout the entire process.

## 3.2 PROPOSED OBJECTIVE FUNCTION

FSTMC aims to fully exploit aleatoric, epistemic, and predictive uncertainties and to minimize, in an end-to-end manner, the composite loss:

$$L = L_{\text{AU}} + L_{\text{EU}} + L_{\text{PU}} + L_{\text{DDC}}. \tag{1}$$

In particular, $L_{\text{AU}}$ leverages aleatoric uncertainty to weight the learning signal and incorporates a log-variance regularizer, thereby mitigating the impact of noisy samples; $L_{\text{EU}}$ penalizes high-uncertainty predictions based on estimates of epistemic uncertainty, alleviating overconfidence and improving generalization stability; $L_{\text{PU}}$ uses predictive uncertainty as weights to align and calibrate the multi-modal posteriors, easing inter-modal conflicts and increasing fusion confidence to ensure a trustworthy final clustering; $L_{\text{DDC}}$ imposes a discriminative clustering constraint on the uncertainty-aware evidence representation, enhancing inter-cluster separability and intra-cluster compactness.

## 3.3 ALEATORIC UNCERTAINTY LEARNING: $L_{AU}$

Aleatoric Uncertainty is a crucial component of reliability assessment. Previous work such as the D-Net model (Yu et al., 2019), have eliminated aleatoric uncertainty by mapping samples to a latent Gaussian distribution and resampling them, introducing an entropy term to avoid variance collapse; however, its sampling-dependent nature means that each iteration only updates a single point in the distribution, which reduces training efficiency and stability. we employ a VAE-style encoder (Lopez et al., 2020) for feature extraction and quantification of random uncertainty. Unlike standard VAEs, we do not use a VAE decoder or loss function because our goal is not generative modeling, but rather obtaining continuous, structured latent representations and their variance estimates. By using a VAE encoder, we preserve a continuous and well-structured latent space that not only characterizes sample noise using random uncertainty but also smoothly interfaces with downstream evidence fusion modules.

In our method, given an input sample $x_i^m$, its VAE encoding result:

$$p(z_i^m | x_i^m) = \mathcal{N}(z_i^m; \mu_i^m, (\sigma_i^m)^2), \tag{2}$$

where $\mu_i^m$ denotes the mean, which serves as the latent feature representation of the sample, while $(\sigma_i^m)^2$ represents the sample-level aleatoric uncertainty, reflecting the degree of noise of the sample in the latent space. To use the latent representation for clustering, we introduce a set of learnable class prototypes $\{\eta_k\}_k^C$. Based on the class-conditional Gaussian assumption, we obtain the posterior distribution of sample $x_i^m$:

$$p(C = k \mid x_i^m) = \frac{\exp\left(-\frac{1}{2(\sigma_i^m)^2}\|\mu(x_i^m) - \eta_k\|^2\right)}{\sum_{j=1}^{C} \exp\left(-\frac{1}{2(\sigma_i^m)^2}\|\mu(x_i^m) - \eta_j\|^2\right)}. \tag{3}$$

After applying $\ell_2$-normalization to both $\mu(x_i^m)$ and the class prototypes $\{w_k\}$, the resulting posterior distribution is equivalent to a Boltzmann distribution adaptively modulated by the aleatoric uncertainty. Specifically, $(\sigma_i^m)^2$ governs the entropy of the distribution: when $(\sigma_i^m)^2$ is small, the distribution becomes sharp with low entropy, indicating high predictive confidence; conversely, when $(\sigma_i^m)^2$ is large, the distribution becomes smooth with high entropy, reflecting low predictive confidence. Consequently, $(\sigma_i^m)^2$ can be naturally interpreted as a quantitative measure of sample quality and reliability.

To use pseudo labels to guide clustering optimization, we assume that the category label of sample $x_i^m$ is $C^m$, and the loss function is expressed by the following formula:

$$\mathcal{L}_a = \frac{1}{(\sigma^m)^2} \text{CE}\left(C^m, \text{softmax}(\mu(W^m)^\top)\right) + \log(\sigma^m)^2. \tag{4}$$

Where $CE$ represents the cross entropy based on pseudo labels, $W^m = \{\eta_k\}_k^C$, denotes the class prototype matrix of the $m$-th modal. Moreover, the first term $\frac{1}{(\sigma^m)^2}$ amplifies the contribution of low-uncertainty samples, ensuring that high-quality samples dominate the clustering optimization; the second term $\log(\sigma^m)^2$ serves as a regularization, preventing degenerate solutions caused by

overly small or excessively large variance, thereby stabilizing the training process. Finally, the aleatoric uncertainty losses from different modalities are averaged across modals:

$$\mathcal{L}_{AU} = \frac{1}{M} \sum_{m=1}^{M} \mathcal{L}_a. \tag{5}$$

To ensure the reliability of pseudo-label supervision, we filter samples based on the predictive uncertainty $u$ obtained from the fusion, fixing the threshold at $\varepsilon = 0.6$, without additional parameter tuning on different datasets, thus avoiding selection bias caused by repeated parameter tuning on different datasets. This setting also ensures that samples $u < 0.6$ have high confidence and can serve as stable pseudo-label supervision. Specifically, only samples satisfying $u < \varepsilon$ participate in the cross-entropy loss, while filtered high-uncertainty samples are still included in the log-regularization term; their larger variance provides constraints on the latent distribution, preventing scale degradation. This strategy effectively reduces the negative impact of noisy samples and improves training stability.

### 3.4 EPISTEMIC UNCERTAINTY LEARNING: $L_{EU}$

Epistemic uncertainty plays a key role, reflecting the model's confidence in its predictions (Sankararaman & Mahadevan, 2011). Unlike approaches that estimate model uncertainty only at the prediction stage, we incorporate it into the representation learning stage so that it directly guides the optimization of latent spaces. Epistemic uncertainty captures the stability of feature representations under parameter perturbations, where significant fluctuations indicate fragile representations and low confidence in the features.

Traditional Bayesian neural networks require prior conditions and perform variational inference in a high-dimensional parameter space, which is not only computationally expensive but also difficult to approximate. To address this, we introduce Monte Carlo (MC) Dropout as a Bayesian approximation in the latent representation layer. This approach offers the advantage of estimating the model's epistemic uncertainty at a low cost. Through multiple forward propagations, MC-Dropout effectively captures the model's response to different parameter perturbations, thereby quantifying the model's stability. This stability estimation helps us make conservative decisions in regions of high uncertainty, avoiding overly confident judgments on unstable features, thus improving the model's robustness and reliability.

Specifically, we model each parameter as:

$$\theta_i^m = \pi_i^m \cdot z_i^m, \quad z_i^m \sim \text{Bernoulli}(1 - \zeta), \tag{6}$$

with $\pi_i^m$ denoting the learnable weights, $z_{ij}$ the dropout mask, and $\zeta$ the dropout rate. This formulation is equivalent to imposing a Bernoulli sparsity prior. Assuming the sample is represented as $x_i^m$, we perform $T$ random forward propagation to obtain a set of potential representations:

$$\{\mu_t(x_i^m)\}_{t=1}^{T}, \tag{7}$$

and define the epistemic uncertainty as the variance across these samples:

$$\tau(x_i^m) = \frac{1}{T} \sum_{t=1}^{T} \|\mu_t(x_i^m) - \bar{\mu}(x_i^m)\|^2, \quad \bar{\mu}(x_i^m) = \frac{1}{T} \sum_{t=1}^{T} \mu_t(x_i^m). \tag{8}$$

To avoid suppressing the natural level of epistemic uncertainty, our goal is not to force the overall uncertainty to be small, but rather to prevent a few samples from producing excessively large variance across MC samples and destabilizing the latent space. To achieve this, we first normalize the epistemic uncertainty within each mini-batch and apply a one-sided Huber robust penalty only to the positive tail that exceeds the statistical range, while leaving normal fluctuations fully preserved. Let $L_E^{(m)}(x)$ represent the penalty term for the $m$-th mode. For a detailed representation, see Appendix A.2. The overall epistemic uncertainty loss is obtained by averaging across all modalities:

$$L_{EU} = \frac{1}{M} \sum_{m=1}^{M} L_E^{(m)}(x). \tag{9}$$

This design suppresses anomalous instability without violating the expressive function of cognitive uncertainty, thus generating more stable latent representations.

## 3.5 Predictive uncertainty Learning: $L_{RU}$

During the clustering process, assessing the confidence of predictions is also crucial. We introduce the Dirichlet distribution for parameterized modeling based on the DS evidence theory. Compared to Softmax, the confidence generated by the Softplus function is more robust and reliable. Specifically, for the $m$th modal, we first map its latent representation $\{\bar{\mu}(X^m)\}$ to an evidence $\mathbf{e}^m = [e_1^m, \cdots, e_k^m]$, and further define the parameters of the Dirichlet distribution as $\boldsymbol{\alpha}^m = [\alpha_1^m, \cdots, \alpha_k^m]$. Based on this, we can obtain the category's belief mass and overall uncertainty mass:

$$b_k^m = \frac{\alpha_k^m - 1}{S^{(m)}}, \quad u^{(m)} = \frac{K}{S^m}, \tag{10}$$

where $S^m = \sum_{i=1}^K \left(e_i^{(m)} + 1\right) = \sum_{i=1}^K \alpha_i^{(m)}$ represents the Dirichlet intensity,and all $K+1$ mass assignments are not less than zero and jointly satisfy the normalization constraint, and their sum is one:

$$u^m + \sum_{k=1}^C b_k^m = 1. \tag{11}$$

This relationship shows that the amount of category evidence directly affects the prediction results: the more sufficient the category evidence is, the higher the corresponding probability is; conversely, when the overall evidence is weak, the uncertainty of the prediction is significantly enhanced.

The Demster-Schafer theory provides a method for integrating evidence from different sources; the combination process can be found in Appendix A.3. The combination rule has several intuitive properties. When both modals exhibit high uncertainty ($u^a$, $u^b$ are large), the fused confidence decreases, preventing overconfidence. When both modals are confident and consistent, the resulting belief mass is significantly strengthened. If only one modal is reliable, the fusion result primarily follows that modal. When strong conflict arises between two modals, the normalization factor explicitly penalizes contradictory evidence, avoiding erroneous amplification. By recursively extending this binary combination to $M$ modals, the global joint mass can be obtained as:

$$W = W_1 \oplus W_2 \oplus \ldots W_M = \{b_k\}_{k=1}^K, u. \tag{12}$$

In addition,to reduce the negative effect of low-quality modals, we introduce a reliability gate $r^v \in (0, 1]$ derived from aleatoric and epis- temic uncertainty during representation learning, as:

$$r^m = \exp(-(\sigma^m)^2 - \tau^m). \tag{13}$$

This choice avoids introducing additional hyperparameters while assigning equal weight to aleatoric and epistemic uncertainties. The evidence is then discounted based on $r^m$, consistent with "source discounting" in evidence theory, which recalibrates the contribution of each modality before fusion. When a modality has high noise or the model is unstable in that modality, its $r^m$ decreases significantly, weakening its evidence at the decision level and preventing conflict in multimodal fusion. Conversely, for modalities with high credibility, $r^m$ approaches 1, ensuring that their effective information is fully preserved during fusion.

Finally, the clustering results and the associated predictive uncertainty are obtained through evidence fusion. To alleviate conflicts during fusion and to encourage consistency across modalities.Based on the research, we designed a loss:

$$\mathcal{L}_{PU} = \frac{1}{(M-1)} \sum_{p=1}^M (\sum_{q \neq p}^M) \Psi(\eta_n^p, \eta_n^q), \tag{14}$$

where $\Psi(\eta_n^p, \eta_n^q)$ measures the conflict of the $n$-th sample between modalities $p$ and $q$, and $\eta$ is an ordered triplet consisting of a modality point $\mathcal{M}$ and a prior distribution $\rho = (\rho_1, \ldots, \rho_k)^\top$. At initialization, $\rho_i = 1/K$ to ensure balanced priors across clusters. The discrepancy function can be further decomposed into a projection divergence and a determinacy factor, with detailed definitions provided in Appendix A.4.

In the evidence generation phase, we adopt a Deep Divergence-based Clustering (DDC) model (Kampffmeyer et al., 2019) to jointly promote intra-cluster compactness and inter-cluster separability. The overall loss is defined as:

$$L_{\text{DDC}} = L_1 + L_2 + L_3. \tag{15}$$

The detailed formulas for $L_1$, $L_2$, and $L_3$ are given in the Appendix A.5.

### 3.6 Optimization

Our method jointly optimizes each module in an end-to-end framework. It is worth noting that no additional weighted hyperparameters were introduced in the loss function design, thus reducing the overhead of parameter tuning during training while ensuring the stability and robustness of the method. However, there are still hyperparameters in the model that need to be tuned, and these hyperparameters will affect the model's training and the final result. The complete optimization process can be found in Appendix A.6.

Furthermore, to demonstrate the rationality of this method, this paper proposes two theorems:

**Theorem 1**: Minimizing the pseudo-label cross entropy on the high-confidence subset is equivalent to minimizing the upper bound of the ideal clustering objective. The detailed proof process can be found in Appendix A.7.

**Theorem 2**: MC Dropout is equivalent to variational inference that introduces a Bernoulli prior distribution on the model parameters, that is, the training objective of MC Dropout is equivalent to maximizing the lower bound. The proof process can be found in Appendix A.8.

## 4 Experiments

In this section, we comprehensively validate the effectiveness of the proposed method through systematic experiments. Core experiments include clustering results and analysis, convergence analysis, dropout rate parameter analysis, and ablation experiments. To improve reproducibility and understandability, we provide complete implementation details in Appendix A.9, a detailed description of the dataset used in Appendix A.10, a summary of the best existing methods in Appendix A.11, additional clustering visualization results in Appendix A.12, analysis of the dropout rate $\zeta$ parameter in Appendix A.13, and visualization analysis in Appendix A.14.

### 4.1 Clustering Results and Analysis

In this section, we conduct experimental evaluations on four datasets of varying scales and characteristics, and compare the proposed FSTMC with twenty MMC methods. The performance of all methods in terms of ACC and NMI is reported in Table 1. Overall, FSTMC demonstrates significant and stable clustering capability across different datasets, and a more detailed analysis and discussion are provided below.

Compared with single-modality and full-modality clustering methods, FSTMC demonstrates a clear advantage in clustering performance. For example, on the ESP-Game dataset, the proposed FSTMC significantly outperforms the best single-modality and full-modality methods in terms of ACC, with improvements of 22.3% and 35.8%, respectively. This indicates that the proposed FSTMC can effectively integrate complementary information across different modalities while mitigating redundant information within individual modalities, leading to more accurate and robust clustering results.

Compared with traditional MMC methods, FSTMC still maintains a decisive lead. For example, on the Caltech-2V dataset, the proposed FSTMC outperforms the best traditional method by 21.2% in terms of ACC and by 25.3% in terms of NMI. This fully demonstrates its generality and superiority across datasets of different scales and types.

Compared with deep MMC methods, FSTMC consistently maintains a clear advantage. On the Caltech-2V, IAPR, ESP-Game and NUS-Wide datasets, the proposed FSTMC achieves improvements of 4.8%, 0.9%, 4.6%, and 2.7% in terms of ACC over the best deep methods, respectively. This indicates that FSTMC can fully exploit the three types of uncertainty to obtain more reliable and accurate feature representations, thereby achieving better clustering results.

Table 1: Clustering performance on the Caltech-2V, IAPR, ESP-Game, and NUS-Wide datasets in terms of ACC and NMI. In the table, bold values indicate the best results, while underlined values denote the second-best results.

| Methods | Caltech-2V | | IAPR | | ESP-Game | | NUS-Wide | |
| --- | --- | --- | --- | --- | --- | --- | --- | --- |
| | ACC | NMI | ACC | NMI | ACC | NMI | ACC | NMI |
| KM | 41.6 | 30.5 | 38.9 | 17.2 | 48.4 | 33.5 | 26.8 | 16.7 |
| Ncuts (TPAMI'00) | 39.9 | 31.2 | 41.9 | 18.9 | 46.5 | 29.9 | 31.6 | 14.1 |
| AmKM | 46.4 | 31.4 | 40.4 | 17.0 | 34.9 | 20.3 | 26.8 | 15.2 |
| AmNcuts (TPAMI'00) | 42.8 | 25.2 | 42.2 | 18.9 | 33.6 | 18.9 | 30.4 | 16.1 |
| CoregMVSC (NIPS'11) | 49.2 | 39.6 | 35.1 | 18.4 | 40.1 | 28.8 | 26.3 | 16.8 |
| RMKMC (IJCAI'13) | 51.4 | 33.5 | 36.4 | 15.9 | 44.7 | 29.7 | 30.5 | 14.5 |
| SwMC (IJCAI'17) | 34.2 | 26.6 | 30.2 | 23.1 | 43.7 | 44.2 | 12.5 | 15.0 |
| ONMSC (AAAI'20) | 34.2 | 26.6 | 21.6 | 11.1 | 17.1 | 18.1 | 16.9 | 15.3 |
| TBGL (TPAMI'22) | 39.6 | 34.9 | 22.6 | 4.0 | 22.2 | 12.0 | 32.1 | 16.5 |
| MMGC (AAAI'23) | 34.9 | 19.7 | 26.1 | 3.3 | 17.5 | 1.4 | 40.2 | 15.9 |
| EAMC (CVPR'20) | 40.3 | 26.6 | 37.1 | 16.4 | 27.1 | 6.5 | 24.6 | 9.7 |
| DEMC (INS'21) | 37.1 | 27.9 | 30.1 | 13.8 | 35.5 | 21.6 | 21.3 | 11.1 |
| SiMVC (CVPR'21) | 51.1 | 36.9 | 42.7 | 18.5 | 35.3 | 16.2 | 25.7 | 10.2 |
| CoMVC (CVPR'21) | 59.2 | 49.2 | 46.7 | 21.5 | 51.8 | 38.2 | 43.9 | 31.2 |
| MFLVC (CVPR'22) | 61.5 | 53.6 | 47.3 | 22.6 | 52.1 | 39.4 | 45.1 | 33.1 |
| DealMVC (ACM MM'23) | 47.6 | 37.9 | 35.0 | 10.8 | 42.7 | 24.7 | 36.4 | 24.4 |
| ICMVC (AAAI'24) | 49.6 | 37.9 | 37.1 | 16.8 | 45.8 | 29.5 | 34.7 | 19.8 |
| DIVIDE (AAAI'24) | 64.1 | 52.9 | 45.6 | 23.0 | 46.5 | 27.0 | 36.3 | 26.0 |
| MSDIB(AAAI'25) | 67.8 | 55.2 | 50.6 | 26.7 | 66.1 | 46.5 | 48.6 | 33.5 |
| SDCIB(ICML'25) | 67.5 | 59.2 | 52.8 | 28.7 | 61.4 | 44.7 | 46.1 | 31.4 |
| **Ours** | **72.6** | **64.9** | **53.7** | **30.5** | **70.7** | **51.0** | **51.3** | **37.0** |
| **Ours vs Best Compared** | 4.8↑ | 5.7↑ | 0.9↑ | 1.8↑ | 4.6↑ | 4.5↑ | 2.7↑ | 3.5↑ |

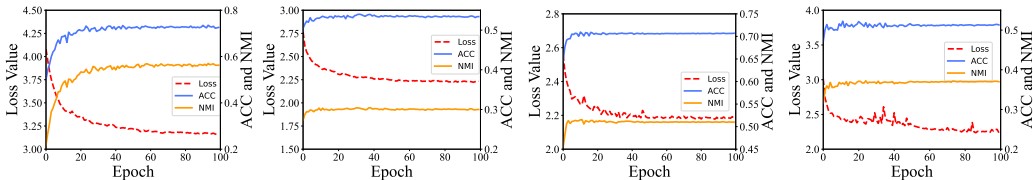

Figure 2: The convergence analysis of FSTMC on the Caltech-2V, IAPR, ESP-Game, and NUS-Wide datasets.

## 4.2 Convergence Analysis

To verify the convergence of FSTMC, we plotted the total loss, ACC, and NMI curves on the Caltech-2V, IAPR, ESP-Game, and NUS-Wide datasets, as shown in Figure 2. The results indicate that the loss function converges to a stable value after approximately 40 epochs, while both ACC and NMI gradually increase and remain at high stable levels. This demonstrates that the proposed FSTMC can converge rapidly within a short training period and consistently reach the optimal solution, validating its reliability and stability.

## 4.3 Ablation Study

To demonstrate the importance of full-stage trustworthiness in MMC, we conduct two types of ablation studies, as shown in Table 2 and Figure 3.

In Table 2, we systematically evaluate the role of different loss components in clustering performance. The results show that $L_{DDC}$ is the core driver of clustering. While relying solely on it can achieve a basic usable clustering structure, overall performance remains limited. When $L_{DDC}$ is combined with a single uncertainty loss, performance improves significantly in most cases, with the

Table 2: The ablation study 1 on the Caltech-2V, IAPR, ESP-Game, and NUS-Wide datasets.

| Methods | Caltech-2V | | IAPR | | ESP-Game | | NUS-Wide | |
|---|---|---|---|---|---|---|---|---|
| | ACC | NMI | ACC | NMI | ACC | NMI | ACC | NMI |
| $L_{DDC}$ | 63.0 | 58.1 | 52.1 | 30.1 | 64.7 | 44.8 | 49.4 | 33.7 |
| $L_{DDC} + L_{AU}$ | 71.9 | 62.3 | **54.7** | **31.3** | 69.7 | 49.9 | 49.8 | 36.8 |
| $L_{DDC} + L_{EU}$ | 67.9 | 59.9 | 52.5 | 30.2 | 66.4 | 47.7 | 48.9 | 34.4 |
| $L_{DDC} + L_{RU}$ | 67.9 | 64.9 | 52.7 | 30.3 | 65.0 | 45.1 | 48.6 | 34.8 |
| FSTMC | **72.6** | **64.9** | 53.7 | 30.5 | **70.7** | **51.0** | **51.3** | **37.0** |

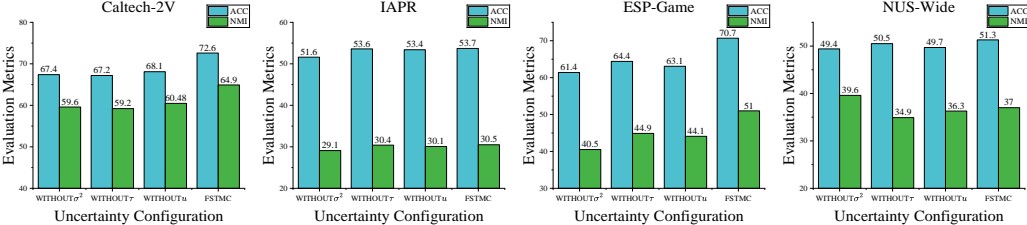

Figure 3: The ablation study 2 on the Caltech-2V, IAPR, ESP-Game, and NUS-Wide datasets.

gains being particularly pronounced on the Caltech-2V dataset. On the IAPR dataset, $L_{DDC} + L_{AU}$ performs slightly better than FSTMC. We speculate that this is due to the small size and high noise content of this dataset. Under these conditions, there may be a conflict between the regularization constraint and cluster separability, resulting in slightly over-regularization or strict pseudo-label filtering in full uncertainty modeling, leading to a single uncertainty loss advantage in some metrics. More notably, on the NUS-Wide dataset, using $L_{DDC}$ alone even outperforms the configuration combined with a single uncertainty loss, suggesting that single uncertainty alone is not sufficient to guarantee performance improvements and may even have negative consequences. Overall, when the three types of uncertainty are introduced simultaneously with $L_{DDC}$, the model achieves optimal or near-optimal results on all datasets. This phenomenon demonstrates that only through joint optimization at all stages can the three types of uncertainty fully leverage their complementary strengths, significantly improving clustering performance and effectively enhancing the credibility and robustness of clustering results.

To further validate the effectiveness of FSTMC, we conducted ablation experiments on the three uncertainty measures, as shown in Figure 4. The results demonstrate that removing either $\sigma^2$ or $\tau$ from the gating coefficient $r$ significantly degrades model performance, highlighting the importance of using high-quality samples and ensuring model stability. Specifically, removing $\sigma^2$ and $\tau$ fails to suppress the interference of low-quality samples on clustering, thus weakening the model's robustness and accuracy. Furthermore, directly using all pseudo-labels without filtering also reduces model performance, indicating that guiding pseudo-label selection based on uncertainty effectively mitigates the negative impact of unreliable labels, thereby improving the accuracy and stability of clustering. By filtering high-confidence pseudo-labels, the model can focus more on high-quality supervisory signals, further enhancing the accuracy and reliability of the clustering results.

## 5 CONCLUSION

In this paper, we propose FSTMC method, the first MMC framework that introduces the notion of full-stage trustworthiness throughout both representation learning and clustering inference. The proposed FSTMC simultaneously quantifies aleatoric, epistemic, and predictive uncertainties. By doing so, it enables multi-level trustworthy constraints and ensures robust optimization during training. Extensive experiments validate the advantages of the proposed FSTMC in terms of clustering performance and stability. Furthermore, this process can be naturally transferred to classification tasks simply by replacing the clustering head and loss with a classification paradigm; future research could further evaluate its performance in cases of missing modalities.

ETHICS STATEMENT

This paper does not involve any potential ethics issues.

REPRODUCIBILITY STATEMENT

We prioritize reproducibility in this research. The main paper clearly explains the proposed method framework, core algorithmic modules, and evaluation process. The appendix provides a complete theoretical justification and the derivation of key propositions. The experiments are based on publicly available datasets and feature a detailed description of the experimental setup. To further ensure reproducibility, we will make the full source code and implementation details publicly available upon acceptance.

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

# A APPENDIX

In the supplemental material:

- **A.1**: The statement of using large language models.
- **A.2**: The specific form of a one-sided Huber robust term.
- **A.3**:Parameter analysis of dropout rate $\zeta$.
- **A.4**: Details of the Discrepancy Function.
- **A.5**: Details of the DDC Loss.
- **A.6**: FSTMC Algorithm.
- **A.7**: Proof of Theorem 1.
- **A.8**: Proof of Theorem 2.
- **A.9**: Implementation details.
- **A.10**: Dataset Descriptions.
- **A.11**: State-of-the-art Methods.
- **A.12**: Visualization Validation.
- **A.13**: Parameter analysis of dropout rate $\zeta$.
- **A.14**: Time complexity analysis.

## A.1 THE STATEMENT OF USING LARGE LANGUAGE MODELS

The authors confirm that the core design and implementation of this study did not rely on any large language model. All work, including the construction of the research framework, methodological design, experimental validation, data processing and analysis, and writing of the paper, was completed independently by the authors.

## A.2 THE SPECIFIC FORM OF A ONE-SIDED HUBER ROBUST TERM

In model training, we use a one-sided Huber robust loss to suppress the impact of unstable samples on the training process and avoid the destruction of the latent space by abnormally high uncertainty. Given the epistemic uncertainty $\tau(x_m^i)$ of a sample in the $m$-th modality, its penalty function is defined as:

$$L_E(\tau(x_i^m)) = \begin{cases} \frac{\tau(x_i^m)^2}{\kappa}, & \text{if } 0 < \tau(x_i^m) \leq \kappa, \\ \tau(x_i^m) - \kappa, & \text{if } \tau(x_i^m) > \kappa. \end{cases} \quad (16)$$

The threshold $\kappa$ is fixed at $0.05$ in the main text and is used to distinguish between "normal fluctuations" and "abnormally high" uncertainty. When $\tau(x_m^i)$ is within the normal range, the loss grows slowly in a quadratic form; after exceeding $\kappa$, it switches to linear growth, thus effectively suppressing the interference of abnormally large variance on training, while not compressing the normal uncertainty expression of the model.

## A.3 EXAMPLES OF THE INTEGRATION OF DEMSTER-SCHAFER THEORY

We combine the DS theory to fuse evidence from different attempts and generate fused belief evidence and uncertainty. Taking modalities 1 and 2 as examples, we can obtain two quality functions: $W_1 = \{b_k^1\}_{k=1}^K, u_1$ and $W_2 = \{b_k^2\}_{k=1}^K, u_2$ through EQ.(10).The joint calculation formula is:

$$W = W_1 \oplus W_2. \quad (17)$$

The detailed calculation rule is defined as follows:

$$b_k = b_k^1 \oplus b_k^2 = \frac{1}{1-C}\left(b_k^1 b_k^2 + b_k^1 u_2 + b_k^2 u_1\right), \quad u = \frac{1}{1-C}u_1 u_2, \quad (18)$$

where $C = \sum_{i \neq j} b_i^1 b_j^2$ is used to measure the inconsistency between the two mass distributions, and the normalization factor $\frac{1}{1-K}$ ensures that the combination result satisfies the probability mass constraint.

### A.4 DETAILS OF THE DISCREPANCY FUNCTION

The discrepancy function is defined as:

$$\Psi_p(\eta_n^p, \eta_n^q) = \frac{1}{2} \sum_{i=1}^{C} |\rho_i^p - \rho_i^q|, \quad \rho_i = \frac{\alpha_i}{S}, \tag{19}$$

and the certainty factor is given by:

$$\Psi_c(\eta_n^p, \eta_n^q) = (1 - u^p)(1 - u^q). \tag{20}$$

### A.5 DETAILS OF THE DDC LOSS

The DDC loss integrates three complementary terms:

$$L_{\text{DDC}} = L_1 + L_2 + L_3, \tag{21}$$

where $L_1$ leverages a generalized Cauchy–Schwartz (CS) divergence to quantify distributional discrepancy and employs a data-driven optimization strategy Kampffmeyer et al. (2019) to enhance sampling effectiveness:

$$L_1 = \frac{1}{c(c-1)} \sum_{i=1}^{c-1} \sum_{j>i} \frac{\delta_i^\top K \delta_j}{\sqrt{(\delta_i^\top K \delta_i)(\delta_j^\top K \delta_j)}}, \tag{22}$$

where $K$ is the Gaussian kernel matrix and $\delta_i$ denotes the $i$-th column of the clustering assignment matrix. $L_2$ introduces a geometric constraint into the CS divergence to align cluster partitions with the vertices of a simplex and avoid degenerate solutions:

$$L_2 = \frac{1}{c(c-1)} \sum_{i=1}^{c-1} \sum_{j>i} \frac{\lambda_i^\top K \lambda_j}{\sqrt{(\lambda_i^\top K \lambda_i)(\lambda_j^\top K \lambda_j)}}, \tag{23}$$

where $\lambda_i$ is the $i$-th column of $B = [B_{pq}] = \exp(-|\alpha_p - e_q|^2)$ and $e_q$ denotes the $q$-th vertex of the simplex. $L_3$ imposes orthogonality among the output vectors to reduce redundancy and improve discriminability:

$$L_3 = \text{triu}(G^\top G), \tag{24}$$

where $\text{triu}(\cdot)$ extracts the upper triangular part.

### A.6 FSTMC ALGORITHM

### A.7 PROOF OF THEOREM 1

Consider the ideal (but unknown) clustering distribution $y_i$ for each sample and the pseudo-label $q_i$ obtained from evidential modeling. On the high-confidence subset $S$, assume that the deviation between $q_i$ and $y_i$ is bounded as $\|q_i - y_i\|_1 \leq \beta$, and that the model prediction is non-degenerate with $\min_k p_{ik}(W) \geq \gamma$.

For a single sample, the difference between the pseudo-label cross-entropy and the ideal clustering objective can be expressed as:

$$\text{CE}(p_i, y_i) - \text{CE}(p_i, q_i) = \sum_k (q_{ik} - y_{ik}) \log \frac{1}{p_{ik}}. \tag{25}$$

Using the deviation bound $\|q_i - y_i\|_1 \leq \beta$ and the non-degeneracy condition $\min_k p_{ik} \geq \gamma$, we obtain:

$$\text{CE}(p_i, y_i) \leq \text{CE}(p_i, q_i) + \beta \log(1/\gamma). \tag{26}$$

---

**Algorithm 1** FSTMC Algorithm

---

**Input**: Multi-modal dataset $\{X^i\}_{i=1}^M$; number of clusters $K$
**Parameter**: Learning rate $\gamma$, dropout rate $\zeta$.
**Output**: The label predictor $C$.

1: Initialize the neural network parameters $\{\theta^i\}_{i=1}^m$.
2: **while** not converge **do**
3:     Probabilistic modeling captures the low-dimensional latent space of data and characterizes aleatoric uncertainty $\sigma^2$ and epistemic uncertainty $\tau$.
4:     Compute the AU loss by Eq. (4) and Eq.(5).
5:     Compute the ED loss by Eq. (8) and Eq.(9).
6:     Input each modal feature into the evidence module to obtain the evidence representation $e$ and predictive uncertainty $u$.
7:     Compute the RU loss by Eq. (14) .
8:     Compute the DDC loss by Eq. (15).
9:     Optimize the overall loss Eq. (1) by adam optimizer and back-propagate loss.
10: **end while**
11: **return** $C$

---

Summing over all samples in $S$ shows that minimizing the pseudo-label cross-entropy on the high-confidence subset is equivalent to minimizing an upper bound of the ideal clustering objective.

Next, consider the gradient of the pseudo-label cross-entropy with respect to the prototype parameter $w_c$:

$$\frac{\partial}{\partial w_c}\text{CE}(p_i(W), q_i) = \sum_{i \in S}(p_{ic} - q_{ic})\tilde{\mu}_i. \tag{27}$$

At convergence, where $p_{ic} \approx q_{ic}$, the optimal prototype direction satisfies:

$$w_c^\star \parallel \sum_{i \in S} q_{ic}\,\tilde{\mu}_i, \tag{28}$$

which demonstrates that each prototype aligns with the pseudo-label weighted class mean direction.

## A.8 PROOF OF THEOREM 2

In our model, dropout is applied at the *feature extraction stage* rather than at the final clustering output layer. Let the parameter vector be $\pi \in \mathbb{R}^d$ and the random mask follow $z \sim \text{Bernoulli}(1 - p)^d$. The effective parameters are then $\tilde{\theta} = \pi \odot z$, and the latent representation of a sample is given by $\mu(x; \tilde{\theta}) = f(x; \pi \odot z)$. From the perspective of variational inference, we can define the variational distribution as:

$$q(\pi, z) = \delta_{\pi=\hat{\pi}} \cdot \prod_j \text{Bernoulli}(z_j; 1 - p). \tag{29}$$

Under this setting, the evidence lower bound (ELBO) consists of two terms: the expected log-likelihood under random masks and an $\ell_2$ regularization term on $\pi$. After simplification, we obtain:

$$\hat{\mathcal{L}}_{\text{drop}}(\hat{\pi}) = \frac{1}{T}\sum_{t=1}^T\left[-\sum_{n=1}^N \log p(y_n \mid f(x_n; \hat{\pi} \odot z^{(t)}))\right] + \lambda\|\hat{\pi}\|_2^2, \tag{30}$$

which is equivalent to the standard dropout training objective. Furthermore, based on $T$ stochastic forward passes, the variance of the latent representations is:

$$\tau(x_i^m) = \frac{1}{T}\sum_{t=1}^T\|\mu_t(x_i^m) - \bar{\mu}(x_i^m)\|^2, \quad \bar{\mu}(x_i^m) = \frac{1}{T}\sum_{t=1}^T \mu_t(x_i^m). \tag{31}$$

which naturally quantifies epistemic uncertainty. Therefore, MC Dropout can be regarded as a Bernoulli variational approximation imposed at the feature extraction layer, enabling epistemic uncertainty to be captured directly during representation learning rather than being deferred to the clustering stage.

## A.9 Implementation details

We build and train the proposed model on PyTorch 1.13.0 (Python 3.8). First, in the latent encoding stage, we employ a VAE-style encoder with heteroscedasticity modeling to generate the mean $\mu$ and variance $\sigma^2$ for each modality, where the latent space dimension is set to 256, the activation function is ReLU, and the variance is guaranteed to be positive via Softplus. To estimate epistemic uncertainty, we introduce MC Dropout in the latent layer and use the dropout rate as an adjustable hyperparameter. During training and inference, we perform multiple random forward propagations (default $T = 5$) to measure the model's sensitivity to parameter perturbations using the variance of the latent representation.

In the evidence generation part, we use a two-layer nonlinear MLP to map the latent features $h_m$ to modal evidence $e_m$. This MLP has a hidden layer dimension of 100 and is equipped with non-linear activation to enhance expressive power. The output layer dimension is consistent with the number of classes, and Softplus activation is used to ensure the non-negativity and numerical stability of the evidence, thus avoiding the "overconfidence" of ReLU and the numerical explosion of the exponential function. Experimental results show that, compared with a simple linear head, this structure makes the DS fusion process more stable and reliable while maintaining extremely low inference cost.

The entire model was trained for 100 epochs using the Adam optimizer with an initial learning rate of 0.001, a batch size of 100, and a pseudo-label filtering threshold of $\epsilon = 0.6$ maintained in all experiments. To improve the stability of the results, we repeated the training for each experiment 20 times and reported the best ACC and NMI obtained at the minimum loss point to avoid getting trapped in local optima.

## A.10 Dataset Descriptions

We selected four widely used benchmark datasets for experimental evaluation. **Caltech-2V** (Fei-Fei et al., 2004) consists of 1,400 images divided into 7 categories, with two modalities, WM and CENTRIST. **IAPR** (Grubinger et al., 2006) includes 7,855 images, each associated with a natural language description, and adopts two modalities, SIFT representation and the BoW model, grouped into 6 categories. **ESP-Game** (Von Ahn & Dabbish, 2004) comprises 11,032 image instances from an online image annotation game, providing two modalities (image descriptions and textual annotations) and organized into 7 categories. **NUS-Wide** (Chua et al., 2009) consists of 20,000 image samples collected from the web, categorized into 8 classes based on image and text modalities.

## A.11 State-of-the-art Methods

To verify the effectiveness of the FSTMC, we introduce twenty MMC approaches as baselines, which overall cover three representative categories of clustering paradigms. It is worth noting that, unlike these methods, the proposed FSTMC can ensure a concise and robust clustering process without any parameter settings, while still delivering stable experimental results. **Single-modality clustering / Full-modality clustering methods:** K-Means (KM) and Normalized Cuts (Ncuts) are typical single-modality clustering methods. In the multi-modal scenario, by concatenating features from different modalities into a single modality for clustering, their extended versions AmKM and AmNcuts can be obtained. **Traditional MMC methods:** CoregMVSC (Kumar et al., 2011), RMKMC (Cai et al., 2013), SwMC (Nie et al., 2017), ONMSC (Zhou et al., 2020), TBGL (Xia et al., 2022), and MMGC (Tan et al., 2023) are six representative traditional MMC methods, each designed from different perspectives to enhance clustering robustness. **Deep MMC methods:** EAMC (Zhou & Shen, 2020), DEMC (Xu et al., 2021), SiMVC and CoMVC (Trosten et al., 2021), MFLVC (Xu et al., 2022), DealMVC (Yang et al., 2023a), ICMVC (Chao et al., 2024), DIVIDE (Lu et al., 2024), MSDIB (Hu et al., 2025a) and SDCIB(Lou et al., 2025b) constitute ten state-of-the-art deep MMC methods, which have achieved competitive performance in recent years.

## A.12 Visualization Validation

To provide a more intuitive illustration of the clustering results, we employ $t$-distributed stochastic neighbor embedding (t-SNE)(Van der Maaten & Hinton, 2008) to visualize the outcomes on Caltech-2V, IAPR, ESP-Game, and NUS-Wide datasets, as shown in Figure 4. In the figure, different colors

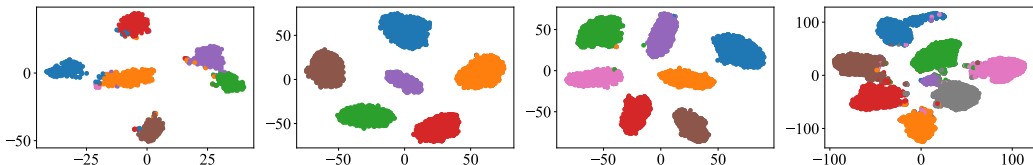

Figure 4: The t-SNE visualization of FSTMC on Caltech-2V, IAPR, ESP-Game, and NUS-Wide dataset.

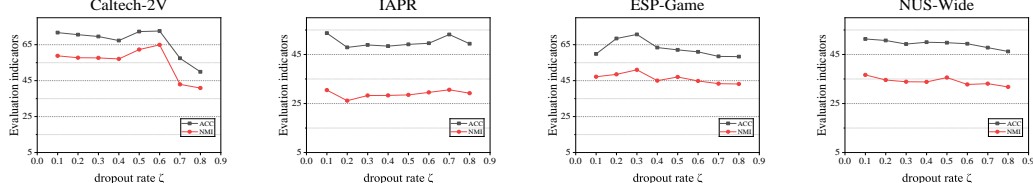

Figure 5: Parameter analysis of Dropout rate $\zeta$ on the Caltech-2V, IAPR, ESP-Game, and NUS-Wide datasets.

correspond to different clusters. It can be observed that, when clustering with the proposed FSTMC, samples from the same category are distributed more compactly and concentrated, while the boundaries between different categories become more distinct, resulting in a clearer and more interpretable overall structure. This demonstrates that the method can effectively capture fine-grained patterns in complex datasets, thereby validating its superior clustering performance.

### A.13 PARAMETER ANALYSIS OF DROPOUT RATE $\zeta$

To validate the stability of our method, we performed a parameter analysis for different dropout rates on all datasets. The results are shown in Figure 5 . Overall, the metrics show little fluctuation with changes in this hyperparameter, with performance only significantly declining at extremely high dropout rates. This phenomenon is consistent with the inherent properties of dropout: its expectation remains stable during training, while the strength of the forward perturbation increases with $\zeta(1 - \zeta)$. Within a typical range of values, this perturbation is insufficient to dominate the uncertainty estimate and therefore does not significantly affect the final results. Only when the dropout rate is too high does the excessive perturbation disrupt representation learning, leading to performance degradation. Notably, on the Caltech-2V dataset, performance degrades significantly when the dropout rate reaches 0.8 or 0.9. Overall, these results indicate that our method is insensitive to this hyperparameter and maintains excellent robustness and stability within a reasonable range.

### A.14 TIME COMPLEXITY ANALYSIS

The main computational cost of FSTMC comes from the "encoder × MC sampling" during training. Assuming the data volume $N$, batch size $B$, number of modalities $M$, forward cost per modality $F_m$, and number of sampling iterations $T$, the complexity per epoch is $O\left(TNB\sum_{m=1}^{M}F_m\right)$. Although the complexity of MC Dropout is approximately linear with $T$, we use a small constant (e.g., $T=5$), so the additional cost compared to baseline methods remains small; and the tradeoff between achieving stable feature representations and improved clustering performance is reasonable and necessary.

