# OpenReview forum: "To the Best of Trust: Full-Stage Trusted Multi-modal Clustering"
_ICLR.cc/2026/Conference — ICLR 2026 Conference Withdrawn Submission_

### Official Review · Reviewer_2wU2 · 2025-10-29

**Soundness:** 2
**Presentation:** 2
**Contribution:** 2
**Rating:** 2
**Confidence:** 3

**Summary:**

This paper proposes FSTMC, a multi-modal clustering method aiming to model and utilize three types of uncertainty (aleatoric $\sigma^2$, epistemic $\tau$, and predictive $u$) in a "full-stage" manner to enhance trustworthiness and robustness1. While the paper addresses an important problem and reports strong empirical results on some metrics, I have reservations regarding its technical soundness, methodological novelty, and experimental validity. The core methodology suffers from a circular dependency, key technical descriptions (especially regarding the VAE) are misleading, and the experimental results (specifically the ablation study) contradict the paper's central thesis. My recommendation is Reject.

**Strengths:**

The paper is relatively clearly written.

**Weaknesses:**

The paper's contribution seems derivative, as it primarily consists of an amalgamation of established methods rather than offering significant innovation.

The core methodology has a severe technical flaw and circular dependency. The authors' proposed "virtuous cycle" is logically circular. Point 1: The Aleatoric Uncertainty Loss ($L_{AU}$) relies on pseudo-labels that are filtered by the predictive uncertainty $u$. Point 2: The predictive uncertainty $u$ is generated via DS evidence fusion. Point 3: This DS fusion process is modulated by a "reliability gate" $r^m$. Point 4: This gate $r^m$ is defined as $r^{m}=\exp(-(\sigma^{m})^{2}-\tau^{m})$, which directly depends on the aleatoric uncertainty $\sigma^2$. In short: the learning of $\sigma^2$ (via $L_{AU}$) depends on $u$, but the calculation of $u$ depends on $\sigma^2$. This is an invalid circular argument. The authors fail to explain how this loop is computationally unrolled, making the core mechanism technically unsound.

The paper's description of its VAE-based method is misleading. The paper claims to "adopt a VAE encoder" for $L_{AU}$. However, the defined loss function $\mathcal{L}_{a}=\frac{1}{(\sigma^{m})^{2}}CE(...) + log(\sigma^{m})^{2}$ is not a VAE loss. A VAE objective must include a reconstruction loss (from a decoder) and a KL divergence term. The paper mentions no decoder, and its loss function is completely different, resembling an aleatoric uncertainty loss for classification (e.g., Kendall & Gal, 2017) rather than a VAE.


The paper makes misleading claims about being "hyperparameter-free". This is entirely false. The pseudo-label threshold $\epsilon=0.6$ is a critical hyperparameter. The MC Dropout rate $\zeta$ is a hyperparameter. The authors even dedicate Section 3.3 and Figure 3 to analyzing it. The number of MC passes T. The 1:1 weighting in $r^m$ is an implicit, fixed hyperparameter choice. These claims are highly misleading.

 The use of $L_{EU}$ is questionable. The paper uses $L_{EU}$ to "penalize high-uncertainty predictions" and suppress "abnormally high uncertainty". This is counter-intuitive to Bayesian principles, which aim to quantify uncertainty, not penalize it. Forcing the model to be confident on difficult samples may reduce trustworthiness, not improve it.

The visualization is insufficient. The t-SNE plot in Figure 5 is standard but uninformative for a paper on uncertainty. The authors should have provided t-SNE plots colored by the learned uncertainty values ($\sigma^2$, $\tau$, or $u$) to visually validate that the model correctly identifies ambiguous or noisy samples.

**Questions:**

I don't have other questions.

---

> ### Author Response · Authors · 2025-11-20
> **Response To Reviewer 2wU2 (Part I)**
>
> We are very grateful for the reviewers' valuable suggestions, especially their comments on motivation, potential impact, and theoretical basis. Regarding the weaknesses and issues raised by the reviewers in our work, we will respond and clarify them one by one below:
>
> ***
> **Weakness1**
> \
> \
> We appreciate the reviewers' valuable comments. Regarding your point that "the contribution mainly comes from the fusion of existing methods," it is true that VAE heteroscedasticity modeling, MC Dropout, and DS evidence fusion are not new technologies we proposed; they already have mature applications in their respective fields. In our work, these techniques are merely means to obtain different types of uncertainty. The real innovation lies in the fact that we are the first to integrate aleatoric, epistemic, and predictive uncertainties throughout the entire process of representation learning and clustering inference, forming a complete and reliable clustering mechanism across all stages. Furthermore, we explain that the "virtuous cycle" in the paper refers to a sequentially executed closed-loop training process, that is, through repeated iterative optimization, continuously improving the model's credibility and robustness through more reliable pseudo-label selection and uncertainty modeling.
> In FSTMC, although there are numerical dependencies between $\sigma^2$, $\tua$, pseudo-labels, and gating in the forward computation, these dependencies are unidirectional, all steps are independent, and the model optimization process does not involve circular dependencies or invalid circular arguments. Thank you again for your feedback. In the revised draft, we have presented this overall modeling approach more clearly and prominently to avoid giving readers the misunderstanding of "simple patchwork".
>
> ***
> **Weakness2**
> \
> \
> Thank you for the reviewers' comments. The paper mentions using a VAE encoder for quantifying random uncertainty, but we did not use the full VAE objective function. VAE-based uncertainty estimation is achieved by learning latent variables. Latent variables can be seen as a representation of the uncertainty of the data. In this study, we adopted a VAE-style encoder, but not the full VAE training objective. Specifically, we did not use the decoder part, therefore there is no reconstruction loss and KL divergence term. VAE-based uncertainty estimation is achieved by learning latent variables. Latent variables can be seen as a representation of the uncertainty of the data. By learning latent variables, the VAE can learn the probability distribution of the data and assign an uncertainty measure to each data point. Our goal is not to generate data through VAE, but to learn the latent representation through the encoder part and compute the random uncertainty $\sigma^2$ of the latent variables. The loss function we defined is not the traditional VAE loss because we did not use the decoder from the generative model, nor did we include reconstruction loss and KL divergence term.
>
>
> ***
> **Weakness3**
> \
> \
> We appreciate the reviewers' valuable comments. Our statement "no hyperparameters" may have been misleading. In fact, we meant "no hyperparameters" only because the loss function does not incorporate any additional weighted hyperparameters (such as $\lambda_1$, $\lambda_2$, etc.). In our model, the loss function is calculated using fixed terms without additional weight adjustments. However, there are indeed hyperparameters in the model that require tuning, and these hyperparameters can affect the model's training and results. We will clarify this point and the meaning of "no hyperparameters" in the revised manuscript.

---

> ### Author Response · Authors · 2025-11-20
> **Response To  Reviewer 2wU2 (Part II)**
>
> **Weakness4**
> \
> \
> We appreciate the reviewers' valuable suggestions. We would like to clarify that the core of the $L_{EU}$design is not to penalize uncertainty itself, but rather to measure the stability of the latent representation by calculating the variance of the latent representation over multiple forward propagations. We impose only a slight constraint on exceptionally high uncertainty through a one-sided Huber robust penalty, preventing a few noisy samples or unstable features from interfering with training, while normal uncertainty fluctuations are preserved. This design ensures that the model can handle samples with high uncertainty while maintaining the stability of the latent representation, thereby improving overall clustering quality and robustness without violating Bayesian principles.
>
> ***
> **Weakness5**
> \
> \
> We appreciate the reviewers' valuable suggestions.We fully agree that in a study focusing on uncertainty, a standard t-SNE plot alone is insufficient to reveal how the model handles ambiguous or noisy samples. A t-SNE visualization colored by the learned uncertainties—such as $\sigma^2$, $\tau$, or $u$—can more intuitively illustrate how different types of uncertainty distribute in the latent space and help verify the model’s ability to distinguish high-noise samples. We are currently supplementing t-SNE visualizations colored by the three types of uncertainty to provide a clearer view of how the model separates samples with varying uncertainty levels in the latent space.
> ***
>
> Finally, we would like to inform the reviewers that the revision and experimental completion of the paper are currently being expedited. We expect to complete all revisions within the next few days and upload the updated PDF version simultaneously. At that time, we will clearly mark the corresponding revisions for each comment in blue in the manuscript and provide a detailed explanation of the updated locations in the rebuttal. We sincerely thank the reviewers for their patience, understanding, and valuable suggestions, and we will continue to do our utmost to ensure that the revised version is more rigorous, complete, and readable.
>
>
> If your concerns have been addressed, Could you please help raise the score. If you have any other concerns, please let us know, and we will try our best to address them. Thanks.

---

> ### Author Response · Authors · 2025-11-25
> **Revisions of our Submission**
>
> Thank you again for your valuable feedback. Our revisions are as follows:
> ***
>
> **Revisions to Weakness 1**
> \
> \
> To avoid misunderstanding, we have removed the term "virtuous cycle" and presented the relationship between the three types of uncertainty in **Section 2 (Three Types of Uncertainty in FSTMC)**.
>
> ***
> **Revisions to Weakness 2**
> \
> \
> In **Section 3.3 (Aleatoric Uncertainty Learning: $L_{AU}$)**, we re-explain the VAE-style encoder.
>
> ***
> **Revisions to Weakness 3**
> \
> \
> In **Section 3.6 (Optimization)**, we provided an explanation of the parameter-free aspect in the loss function.
> ***
> **Revisions to Weakness 4**
> \
> \
> In **Section 3.4 (Epistemic uncertainty learning: $L_{EU}$)**, we restate the core of the $L_{EU}$ design.
>
> ***
> **Revisions to Weakness 5**
> \
> \
> We apologize that we are not yet able to provide ideal visualizations in the current version. We have provided a detailed explanation of the three types of uncertainty and their interrelationships through textual descriptions. We are further optimizing the relevant sections and plan to provide clearer and more expressive diagrams in the final version.

---

### Official Review · Reviewer_3Z3a · 2025-10-30

**Soundness:** 2
**Presentation:** 1
**Contribution:** 2
**Rating:** 2
**Confidence:** 3

**Summary:**

This paper proposes a method for clustering data that comes in different modalities. The method is intended to account for aleatoric and epistemic uncertainty in each modality, and to combine modalities in a principled way while accounting for such uncertainty. The key steps are roughly as follows:
1. Each modality is first encoded using a VAE, i.e. each sample of each modality is mapped to a Gaussian of a certain mean (the feature representation) and variance (interpreted as the "aleatoric uncertainty"). A loss $L_{AU}$ is used that is essentially a kind of pseudo-clustering objective that tries to regularize appropriately by the AU.
2. The variance in the feature representation across multiple MC-dropout samples is interpreted as the "epistemic uncertainty". A loss $L_{EU}$ (based on a Huber loss) is used to penalize feature representations that are excessively unstable.
3. After such encoding, a Deep Divergence Clustering (DDC) approach is used to "soft cluster" the embeddings separately per modality. This results in $M$ soft clusterings, where $M$ is the number of modalities.
4. Each soft clustering generated by DDC (one per modality) is mapped to an "evidence" vector. These are then combined across modalities into a Dirichlet distribution using Dempster-Shafer theory. This essentially yields the final multimodal clustering.

Experiments on various multimodal datasets indicate that this method performs well as compared to prior methods when measured in terms of clustering accuracy and normalized mutual information.

**Strengths:**

Using Dempster-Shafer theory to combine information across modalities (by viewing each modality's clustering as providing one form of "evidence") seems like an interesting and novel idea. The empirical performance of the method also seems quite favorable at least on the datasets tested.

**Weaknesses:**

The paper is unfortunately quite challenging to follow and suffers from a lack of conceptual clarity, with the presentation emphasizing engineering complexity over scientific insight. It combines a variety of techniques (VAEs, pseudolabeling, MC-dropout, Dirichlet evidence modeling, reliability gating, Dempster–Shafer fusion, Deep Discriminative Clustering, and many more) without a clear unifying principle or justification. The roles and interactions of these components are insufficiently motivated, and mechanisms such as the reliability gate appear somewhat ad hoc. Moreover, it is not always clear what models are being used in different components and what the learnable parameters of the whole system are.

Overall, while the empirical results are strong, the paper does not manage to communicate the essential scientific insights in an accessible way, and I cannot recommend acceptance in this current state. In particular, the work would benefit from a clearer formulation of the underlying modeling assumptions, a simplified core mechanism, and a more focused and systematic presentation that keeps unnecessary complexity to a minimum. Also, the paper needs to provide a much more detailed comparison with related work.

**Questions:**

One of the biggest things missing from the paper is a clear discussion of the modeling assumptions and the meaning of the various kinds of uncertainty. Specifically:
- Aleatoric uncertainty is always a property of the true data distribution, not of the method. What links the estimated aleatoric uncertainty (i.e. the variance of the VAE output) to true aleatoric uncertainty (e.g. noise in the input, etc)? Why would noisy inputs have high variance? This requires a discussion of the modeling assumptions and what the "true" aleatoric uncertainty is under those assumptions.
- Epistemic uncertainty being estimated via MC-Dropout variance is also insufficiently motivated. The original MC-Dropout line of work considered Bayesian models, where there is a full posterior over predictions. As far as I can tell, this work does not use a Bayesian network for the VAE. So what is the form of epistemic uncertainty that is being modeled? Why is MC-Dropout a useful proxy for it? (I suspect a more useful term for it is merely "representation stability", although I am again not sure why we care about stability under random perturbations induced by MC-Dropout.)
- Typically, predictive uncertainty is a combination of aleatoric and epistemic uncertainty. However, in this paper it seems to be an orthogonal kind of uncertainty, and in fact is never formally defined. Can the authors provide a formal definition and discuss what exactly is being captured by it, and how it relates to AU and EU?

Some other questions:
- What exactly is the $L_{E}^{(m)}(x)$ term used in Eq 9? It is never formally defined. How does it relate to $\tau(x)$ defined in Eq 8?
- In Eq 10 and 11 and other places, what is $C$? Is it the same as $K$, the number of clusters? Is the number of clusters known a priori?
- What exactly are the different models and the associated learnable parameters in the whole system? The loss function is simply written down as an algebraic expression without making it clear what the parameters are.
- Minor: Why does the predictive uncertainty component of the loss (Eq 16) have subscript RU instead of PU?

---

> ### Author Response · Authors · 2025-11-20
> **Response To Reviewer 3Z3a (Part I)**
>
> We are very grateful for the reviewers' valuable suggestions, especially their comments on motivation, potential impact, and theoretical basis. Regarding the weaknesses and issues raised by the reviewers in our work, we will respond and clarify them one by one below:
> ***
> **Weakness1**
> \
> \
> Thank you for your feedback. The core of FSTMC is not simply a mechanical assembly of modules like VAE, but rather a tool for quantifying and propagating uncertainty. Uncertainty learning is central to the entire learning and clustering process: the data layer uses aleatoric uncertainty to suppress the impact of observational noise on the representation; the model layer uses epistemic uncertainty to constrain stability against random perturbations; and the decision layer uses predictive uncertainty to reduce cross-modal conflicts and characterize the credibility of the results. Therefore, each component has a clear division of labor and interacts with each other, rather than being arbitrarily pieced together. The so-called "reliability gate" is not a temporary choice, but rather originates from source discounting in evidence theory: mapping the aleatoric and epistemic uncertainties of each modality into discount weights, used to lower the evidence strength of low-credibility modalities or increase their dispersion, preventing them from amplifying conflicts during fusion.
> ***
> **Weakness2**
> \
> \
> We thank the reviewers for their valuable feedback on the readability and presentation of the paper. We understand that the current version is still not clear enough in terms of modeling assumptions, concept definitions, and mechanism delineation. To address this, we will provide independent and clear definitions for the three types of uncertainty-related concepts in the revised manuscript, and re-examine the core mechanisms, presenting them with a more intuitive structural diagram and clearer logic. We will also supplement the manuscript with more comprehensive comparisons with related works to highlight the innovations and methodological differences of this paper.
>
> ***
> **Question1**
> \
> \
> Thank you for your question. We assume that the observation $x$ is represented by the latent variable $z$ and the observation noise, where the noise constitutes the "aleatoric" uncertainty. The encoder outputs the sample-level posterior:
> $$
> q(z \mid x) = \mathcal{N}(\mu(x), \Sigma_z(x)),
> $$
> where $\Sigma_z(x)$ represents this unavoidable noise in the observation data. The encoder "expresses" the presence of this noise by increasing the variance, thereby avoiding modeling the noise incorrectly as a reliable signal. When the input noise is large or information is missing, the uncertainty of the latent features increases. The encoder avoids adding too much noise to the latent representation by increasing the variance $\Sigma_z(x)$, thus ensuring that the noise is not treated as a deterministic signal. Specifically, $\Sigma_z(x)$ represents the impact of noise, i.e., the noise of the sample/modality, on the latent representation; the variance of the VAE output $\sigma^2(x)$ is a proxy for this noise, which is actually synchronized with the "real" aleatoric uncertainty and can capture the unavoidable random variations in the input data caused by noise or distortion.
>
>
> ***
> **Question2**
> \
> \
> Thank you for your question. We use MC-Dropout in VAE to estimate epistemic uncertainty primarily because it provides a low-cost way to quantify the model's stability to input perturbations. MC-Dropout captures the model's sensitivity to random perturbations by performing multiple forward propagations on the same input, thus effectively estimating epistemic uncertainty. Although the initial work on MC-Dropout was based on Bayesian neural networks and relied on complete posterior inference, in VAE we did not use a Bayesian network. Instead, we computed the variance of the output through multiple forward propagations, which reflects the model's uncertainty under specific inputs, especially when the data is sparse. We care about this stability estimation because it helps us make conservative judgments about the model in regions of high uncertainty, avoiding overly confident decisions on unstable features, thereby improving overall robustness and reliability.

---

> ### Author Response · Authors · 2025-11-20
> **Response To Reviewer 3Z3a (Part II)**
>
> **Question3**
> \
> \
> Thank you for the reviewers' questions. In this paper, predictive uncertainty refers to the decision uncertainty resulting from evidence fusion. It does not directly stem from a simple combination of aleatoric or epistemic uncertainty, but rather is based on the result of fusing multiple modalities of evidence. We parameterize the evidence using the Dirichlet distribution and calculate the entropy and conflict degree of the posterior distribution using Dempster-Shafer fusion theory to measure the uncertainty of the final decision. Aleatoric and epistemic uncertainties act as gating mechanisms to control the strength of evidence in individual modalities, indirectly determining the final predictive uncertainty. In short, predictive uncertainty is a comprehensive manifestation of aleatoric and epistemic uncertainties at the decision-making level, reflecting the credibility of the final decision by measuring the consistency and stability of the fused evidence.
> ***
>
> **Other Question**
> \
> \
> Thank you for the reviewers' questions. We have made the following detailed explanations and revisions to these questions:
>
> 1. Regarding the $L_E^{(m)}(x)$ term in Equation 9: We will explicitly describe $L_E^{(m)}(x)$ in Equation 9. This is our penalty term for epistemic uncertainty, specifically used to adjust the model's output in unstable regions via Huber loss. We will explain this loss term in detail in the main text. The relationship between Equations 9 and 8 is that Equation 8 defines the variance $\tau(x_m^i)$ of epistemic uncertainty, while Equation 9 represents the robustness penalty applied to this uncertainty, primarily a one-sided penalty for excessively high uncertainty.
>
> 2. Regarding the definition of $C$: In Equations 10 and 11, $C$ represents the number of categories (i.e., the number of clusters). For multimodal clustering tasks, the number of clusters $K$ needs to be predetermined based on the actual situation of the dataset. This is usually selected experimentally or set based on domain knowledge. In this paper, $C$ is equivalent to $K$ and is used as a parameter in the Dirichlet distribution and Dempster-Shafer fusion process in the formula, representing the number of categories.
>
> 3. Regarding the model and learnable parameters: Our framework involves multiple modules and parameters. The main models include the VAE encoder, MC-Dropout layer, evidence generation module, and Dempster-Shafer fusion module for clustering. Each module has different learnable parameters. For example, the VAE module has its hidden layer and weight parameters, the MC-Dropout module is used to estimate the uncertainty of the model, and the evidence generation module generates fusion signals by learning the relationships between modalities. These parameters are optimized during training and affect the final clustering results.
>
> 4. Regarding the RU and PU subscripts in Equation 16: The subscript of RU in Equation 16 is incorrect. It should be PU (i.e., predictive uncertainty) instead of RU. We will standardize the notation in the revised draft to ensure that PU is consistently used to represent predictive uncertainty.
>
> ***
>
>
> Finally, we would like to inform the reviewers that the revision and experimental completion of the paper are currently being expedited. We expect to complete all revisions within the next few days and upload the updated PDF version simultaneously. At that time, we will clearly mark the corresponding revisions for each comment in blue in the manuscript and provide a detailed explanation of the updated locations in the rebuttal. We sincerely thank the reviewers for their patience, understanding, and valuable suggestions, and we will continue to do our utmost to ensure that the revised version is more rigorous, complete, and readable.
>
> If your concerns have been addressed, Could you please help raise the score. If you have any other concerns, please let us know, and we will try our best to address them. Thanks.

---

> ### Author Response · Authors · 2025-11-25
> **Revisions of our Submission**
>
> Thank you again for your valuable feedback. Our revisions are as follows:
> ***
> **Revisions to Weakness 1**
> \
> \
> 1.In **Section 3.5 (Predictive uncertainty learning: $L_{RU}$)**, we explained mechanisms such as reliability gating.
>
> 2.In **Appendix A.9 (Implementation details)**, we explain the parameters used in our model.
> ***
> **Revisions to Weakness 2**
> \
> \
> 1. In **Section 2 (Three Types of Uncertainty in FSTMC)**, we focus on the three types of uncertainty and their relationships.
>
> 2. In **Appendix A.9 (Implementation details)**, we explain the architecture of the model.
>
> ***
> **Revisions to Question1**
> \
> \
> In **Section 2 (Three Types of Uncertainty in FSTMC) and Section 3.3 (Aleatoric Uncertainty Learning: $L_{AU}$)**, we explained the relationship between noise and aleatoric uncertainty.
>
> ***
> **Revisions to Question2**
> \
> \
> In **Section 3.3 (Epistemic Uncertainty Learning: $L_{EU}$)**, we further elaborate on MC-Dropout.
>
> **Revisions to Question3**
> \
> \
> In **Section 2 (Three Types of Uncertainty in FSTMC)**, we introduced the definition of predictive uncertainty and its relationship with Aleatoric Uncertainty and Epistemic Uncertainty.
>
>
>
>
>
> **Revisions to Other Question**
> \
> \
> 1.We have corrected the issues in these formulas, as shown in **Section 3.4, Appendix A.2, Formulas 9, 10 and 14**.
>
> 2.In **Appendix A.9 (Implementation details)**, we explain and introduce the model and parameters of this framework.

---

> > ### Comment · Reviewer_3Z3a · 2025-11-26
> > **Still missing clarity**
> >
> > Thank you for your detailed responses. I still feel that there are some important elements missing from the conceptual picture. For example:
> > - "The data layer uses aleatoric uncertainty to suppress the impact of observational noise on the representation" --- I am not sure what it means to "use" aleatoric uncertainty in this way. I believe that what is happening in just that a VAE is being used to compute the latent representation, and we simply assume that the VAE's variance corresponds to aleatoric uncertainty. But has this been validated at all? For example, you say "$\Sigma_z(x)$ represents this unavoidable noise in the observation data". What does "represents" mean here? Is the VAE variance actually high on inputs with high noise? Otherwise the VAE's role is just an optimistic hypothesis.
> > - "The model layer uses epistemic uncertainty to constrain stability against random perturbations" --- again, it is not clear why this is "epistemic uncertainty" (epistemic in what sense exactly?). Rather, it seems to simply be some kind of stability to random MC-Dropout perturbations, and it is not really clear why this is meaningful. Again, its role needs to be validated.
> >
> > In short it still seems that the overall mechanism is composed of various pieces that are _hopefully_ performing certain roles, but (a) the role is not always well-defined (e.g., what is true AU, true EU), and (b) it is not clear that they are _actually_ performing those roles.
> >
> > Overall I am leaning towards increasing my score to 4 but no higher.

---

### Official Review · Reviewer_bv6T · 2025-10-31

**Soundness:** 2
**Presentation:** 3
**Contribution:** 2
**Rating:** 4
**Confidence:** 3

**Summary:**

This paper addresses a critical yet often overlooked issue in the field of multimodal clustering (MMC): end-to-end trustworthiness. The authors propose a unified framework that models three core types of uncertainty, such as aleatoric, epistemic, and predictive, and integrates them coherently throughout both the representation learning and clustering optimization stages. This is a novel and conceptually meaningful approach that carries substantial theoretical and practical significance. From a technical standpoint, the paper skillfully combines Variational Autoencoders (VAE), MC Dropout, and Dempster–Shafer Theory, providing a sound modeling basis for the quantification and utilization of each type of uncertainty.

**Strengths:**

- The overall structure and organization of the paper are well-designed and logically presented.
- The motivation is clearly stated and easy to follow, effectively highlighting the significance of the work.
- The figures are visually appealing, with color schemes that enhance readability and overall presentation quality.
- The experimental section is thorough and well-executed, offering convincing empirical support for the proposed framework.

**Weaknesses:**

- The proposed model integrates multiple components (e.g., VAE and MC Dropout, which requires T forward passes), implying that its training cost may be higher than that of some baseline methods.
- In the description of L_EU, the paper mentions a “one-sided Huber robust”, but its explicit mathematical formulation is not provided in the main text. It would be beneficial to include a clear and intuitive explanation or a simplified expression of this penalty term in the main body of the paper to help readers better understand its role and behavior.
- It would be highly beneficial to include a visualization of the different types of uncertainties—namely aleatoric, epistemic, and predictive—to illustrate their respective values across representative samples. Such a figure would help readers intuitively understand how the model distinguishes among these uncertainties and why it assigns specific uncertainty levels to different modalities or data instances.

**Questions:**

- In Section 2.5, the paper states that “…we first map its latent representation … to an evidence e^m….” This evidence serves as the input to the entire evidence fusion module. However, the specific form of this mapping function is not clearly described. Is it implemented as a simple linear layer, or as a MLP with a nonlinear activation function (e.g., Softplus, to ensure the evidence values remain positive)?
- The paper empirically demonstrates that the joint modeling of all three types of uncertainty yields the best performance. However, it lacks a deeper theoretical or intuitive analysis explaining why these three components are indispensable. Why are all three uncertainties necessary?
- What are the potential risks if only aleatoric uncertainty is considered?
- Conversely, what issues would arise if only epistemic uncertainty were taken into account?

---

> ### Author Response · Authors · 2025-11-20
> **Response To Reviewer bv6T (Part I)**
>
> We greatly appreciate the reviewer’s valuable suggestions, especially the comments on motivation, potential impact, and theoretical foundations. Regarding the weaknesses and issues raised by the reviewer in our work, we will respond and clarify each point as follows:
> ***
> **Weakness1**
> \
> \
> Thank you for your valuable feedback. We acknowledge that introducing MC Dropout introduces additional computation that grows approximately linearly with the number of samples $T$; however, our method achieves good performance with only a small constant-order sampling (e.g., $T = 5$), thus keeping the increase relative to the baseline within a finite factor. Considering the significant improvements in sample-level epistemic estimation, fusion stability, and calibration it brings, we believe this cost-benefit trade-off is reasonable and necessary.
>
> ***
> **Weakness2**
> \
> \
> Thank you for your valuable feedback. We use a one-sided Huber robust loss to suppress excessive epistemic uncertainty caused by model parameter fluctuations and to prevent abnormally high uncertainty from excessively interfering with the training process. Specifically, the one-sided Huber loss penalizes epistemic uncertainty only when the uncertainty value exceeds a certain threshold, thus avoiding over-penalizing normal fluctuations. In our implementation, the model uncertainty $\tau(x_m^i)$ is represented as the variance of sample $x_m^i$ over multiple forward propagations. For the epistemic uncertainty penalty, we use the following form:
> $$
> \begin{aligned} &\text{When } 0<\tau(x_m^i)\le 0.05,
> \quad L_{EU}^{(m)}(x_m^i)=\frac{\tau(x_m^i)^2}{0.05};\\ &\text{When } \tau(x_m^i)>0.05,
> \quad L_{EU}^{(m)}(x_m^i)=\tau(x_m^i)-0.05. \end{aligned}
>  $$
> In this expression, $\tau(x_m^i)$ is the uncertainty of the model with respect to sample $x_m^i$, and $0.05$ is the threshold we set to control the rate of loss growth. This hyperparameter’s effect is that when the uncertainty is less than $0.05$, the loss grows quadratically; when the uncertainty is greater than $0.05$, the loss grows linearly. We employ this mechanism to suppress the excessive influence of unstable samples on model training while avoiding over-penalizing normal fluctuations. We will explain this further in the revised draft to help readers better understand its role.
> ***
> **Weakness3**
> \
> \
> Thank you for your valuable feedback. We fully agree with your point: simultaneously displaying the distribution of the three types of uncertainty—chance uncertainty, cognitive uncertainty, and predictive uncertainty—on representative samples would greatly enhance the reader's intuitive understanding of the model's behavior. Such visualizations would not only clearly show the differences between the various uncertainties in the latent space but also help explain why the model assigns different levels of uncertainty to specific modalities or samples. Therefore, we are supplementing the visualizations of the three types of uncertainty and will present these results in the revised version to further enhance the paper's interpretability and readability.
>
>
> ***
> **Question1**
> \
> \
> Thank you for your question. We employ a two-layer nonlinear MLP as the evidence mapping $h^{m}\to e^{m}$, with the hidden layer dimension set to $100$ to facilitate the calculation of DDC loss. Softplus activation is used at the output to ensure that the evidence is non-negative, differentiable everywhere, and numerically stable, thus avoiding the overconfidence of ReLU and the exponential explosion of $exp(\cdot)$. Nonlinear activation is used in the intermediate layers, resulting in an overall nonlinear mapping. Compared to a simple linear head, this configuration demonstrates better DS fusion stability in our experiments, while the inference stage overhead remains almost unchanged.
>
>
> ***
> **Question2**
> \
> \
> Thank you for your question. In FSTMC, three types of uncertainty interact and are indispensable: aleatoric uncertainty captures "uncontrollable" factors such as observation noise and sample/perspective quality; epistemic uncertainty reflects the "instability" of the model itself, which, if left uncontrolled, will result in "unfounded high confidence" at points of data sparsity or distribution shift, worsening calibration and risk identification; predictive uncertainty pertains to the decision-making side, fusing evidence from various modalities using data fusion (DS) and using distribution dispersion and conflict to characterize "whether this fusion result is usable," directly used for pseudo-label gating and risk control. In other words, the first two (data and model) provide "evidence quality," while the last one imposes quality constraints on the final probability and pseudo-label selection. These three are complementary at the hierarchical level and interconnected at the mechanistic level; joint modeling will simultaneously improve both performance and credibility.

---

> ### Author Response · Authors · 2025-11-20
> **Response To Reviewer bv6T (Part II)**
>
> ***
> **Question3**
> \
> \
> Thank you for your question. Measuring only epistemic uncertainty is equivalent to simplifying the problem to "noise reduction." This ignores the epistemic instability introduced by model parameters, allowing the algorithm to still give high-confidence outputs in regions where the model is not yet reliable, such as when the sample size is small or the distribution is biased. In evidence fusion, it is not possible to reduce the weight of "low-noise but unlearned" modes, which can easily amplify mode conflicts and thus reduce the accuracy of the results.
>
> ***
> **Question4**
> \
> \
> Thank you for your question. If we only measure epistemic uncertainty and ignore the data uncertainty caused by low sample quality and noise, we cannot effectively suppress noisy data. This leads to the introduction of continuous noise gradients into representation learning, causing unreliable evidence to be erroneously amplified, resulting in increased conflict in DS fusion and weakening the accuracy and stability of the results.
>
> ***
>
> Finally, we would like to inform the reviewers that the revision and experimental completion of the paper are currently being expedited. We expect to complete all revisions within the next few days and upload the updated PDF version simultaneously. At that time, we will clearly mark the corresponding revisions for each comment in blue in the manuscript and provide a detailed explanation of the updated locations in the rebuttal. We sincerely thank the reviewers for their patience, understanding, and valuable suggestions, and we will continue to do our utmost to ensure that the revised version is more rigorous, complete, and readable.
>
>
> If your concerns have been addressed, Could you please help raise the score. If you have any other concerns, please let us know, and we will try our best to address them. Thanks.

---

> ### Author Response · Authors · 2025-11-25
> **Revisions of our Submission**
>
> Thank you again for your valuable feedback. Our revisions are as follows:
> ***
> **Revisions to Weakness 1**
> \
> \
> In **Appendix A.14 (Time complexity analysis)**, we performed a time complexity analysis.
>
> ***
> **Revisions to Weakness 2**
> \
> \
> In **Appendix A.2 (The specific form of a one-sided Huber robust term)**, we explain the specific form of a one-sided Huber robust term.
>
> ***
> **Revisions to Weakness 3**
> \
> \
> In the current revised version, we were unable to present these visualizations, but we have made every effort to explain the three types of uncertainty and their interrelationships in detail through textual descriptions. We promise that in future work, we will continue to optimize the visualizations and provide clearer and more intuitive diagrams in the final version to further enhance the readability and interpretability of the paper.
>
>
> ***
> **Revisions to Question1**
> \
> \
> In **Appendix A.9 (Implementation details)**, we explain the specific form of the mapping function.
>
> ***
> **Revisions to Question2, Question3 and Question4**
> \
> \
> In **Section 2 (Three Types of Uncertainty in FSTMC)**, we defined these three types of uncertainty and explained why these three components are indispensable.

---

### Official Review · Reviewer_Zs29 · 2025-10-31

**Soundness:** 3
**Presentation:** 3
**Contribution:** 3
**Rating:** 8
**Confidence:** 5

**Summary:**

This paper proposes a novel full-stage trusted multi-modal clustering method, namely FSTMC, a framework that integrates three types of uncertainty, including aleatoric, epistemic, and predictive, throughout the entire pipeline of representation learning, evidence fusion, and pseudo-label optimization, thus establishing an end-to-end trustworthy clustering paradigm. Generally, the proposed FSTMC method gives some new insights, theoretically well-founded, and extensive experimental results demonstrates significant performance improvements across multiple benchmark datasets.

**Strengths:**

1. This paper proposes the novel “Full-Stage Trusted” paradigm, extending uncertainty learning from the representation learning stage to the clustering and pseudo-label optimization stages, thereby achieving an end-to-end trustworthy constraint.

2. The study introduces model uncertainty into the variational autoencoder structure, enabling trustworthy modeling at the latent representation level and enhancing the reliability of feature learning.

3. The model shows insensitivity to dropout rate parameter setting, maintaining stable convergence and high clustering accuracy across different configurations.

4. Extensive experiments on multiple large-scale multi-modal datasets demonstrate that the proposed method significantly outperforms existing approaches in terms of clustering accuracy, validating its superior performance and broad applicability.

5. Two types of ablation studies are conducted to comprehensively demonstrate the advantages of computing and utilizing uncertainty within the framework.

**Weaknesses:**

1.	The selection basis for the pseudo-label confidence threshold \varepsilon is not explained.
2.	The paper does not explore the model’s performance under modality-missing scenarios, which limits understanding of its behavior in incomplete multi-modal environments.
3.	Some symbols are not consistently defined upon their first appearance, affecting the readability. Also, the text in some figure annotations (e.g., Figure 3) is too small, making it difficult to read.

**Questions:**

1.	How is the pseudo-label confidence threshold \varepsilon selected, and can a detailed description or experiment be added in the paper to demonstrate the rationality of its choice?
2.	Is uncertainty estimation performed within each modality, and what are the advantages of doing so?
3.	Does any correlation exist among the three types of uncertainty, and how can it be interpreted?
4.	Can this framework be extended to other tasks, such as classification, and if so, how should it be implemented?

---

> ### Author Response · Authors · 2025-11-20
> **Response To Reviewer Zs29**
>
> We greatly appreciate the reviewer’s valuable suggestions, especially the comments on motivation, potential impact, and theoretical foundations. Regarding the weaknesses and issues raised by the reviewer in our work, we will respond and clarify each point as follows:
> ***
> **Weakness1 and Question 1**
> \
> \
> Thank you for your valuable opinion and question. In this paper, we fixed $\varepsilon$ to 0.6 for two main reasons: firstly, for fairness and reproducibility, to avoid bias introduced by tuning the parameter on each dataset; secondly, based on our uncertainty measure $u \in [0,1]$，$u<0.6$ corresponds to "high confidence, evenly distributed samples", which helps to ensure the quality of pseudo-labels while suppressing the negative effects of incorrect labels, without significantly reducing the number of adopted samples. To avoid excessive fine-tuning of the threshold in the final manuscript, we chose this conservative and general fixed value rather than adjusting it for each dataset.
>
> ***
> **Weakness2**
> \
> \
> Thank you for your valuable opinion. We did not discuss the performance in the scenario of missing modalities. This will be addressed as a future research direction. In future work, we plan to evaluate the model’s performance in incomplete multimodal environments to further validate its robustness and adaptability.
>
> ***
> **Weakness3**
> \
> \
> Thank you for your valuable feedback. We will carefully review and ensure that symbols are clearly defined when they first appear to improve readability. Additionally, we will adjust the text size in the figure annotations to ensure that the charts are clearer and easier to read.
>
>
> ***
> **Question2**
> \
> \
> Thank you for your question. We independently estimate uncertainty within each modality: using VAE  to obtain aleatoric uncertainty, and the variance of multiple forward passes with MC Dropout to obtain epistemic uncertainty; then, the soft labels of each modality are mapped to Dirichlet evidence and fused via DS to obtain predictive uncertainty. This approach has three benefits:
> 1. Explicitly modeling aleatoric/epistemic uncertainty for different modalities, reducing the interference of sample noise and parameter uncertainty on feature representation;
> 2. Calculating the predictive uncertainty for each modality, explicitly reducing cross-modal conflicts in DS fusion;
> 3. In cases of modal quality imbalance, the decision is led by the high-confidence modality, resulting in more reliable clustering results and significantly improving overall robustness.
> ***
>
>
> ***
> **Question3**
> \
> \
> Thank you for your question. The three uncertainties interact and are all necessary: aleatoric captures uncontrollable factors like observational noise and sample/view quality; epistemic reflects the model's own uncertainty which, if not controlled, could lead to unjustified high confidence in sparse data or shifted distributions, worsening calibration and risk identification; predictive belongs to decision-side uncertainty, which fuses evidence from multiple modalities using DS, with the distribution's spread and conflict quantifying "how usable this fusion result is" directly used for pseudo-label gating and risk control. In other words, the first two provide evidence quality, while the last applies this quality constraint to the final probabilities and pseudo-label selection, with the three complementing each other at the hierarchical level and working together mechanistically, resulting in both performance and trustworthiness improvements when jointly modeled.
>
> ***
> **Question4**
> \
> \
> Thank you for your question. This framework can be extended to other tasks, such as classification. Specifically, we keep the feature extraction and DS fusion stages intact, simply replacing the clustering head with a classification head and replacing the DDC loss with a conventional classification loss, thus fully reusing our "full-stage trustworthiness" approach for clustering in classification tasks.
>
> ***
>
> Finally, we would like to inform the reviewers that the revision and experimental completion of the paper are currently being expedited. We expect to complete all revisions within the next few days and upload the updated PDF version simultaneously. At that time, we will clearly mark the corresponding revisions for each comment in blue in the manuscript and provide a detailed explanation of the updated locations in the rebuttal. We sincerely thank the reviewers for their patience, understanding, and valuable suggestions, and we will continue to do our utmost to ensure that the revised version is more rigorous, complete, and readable.
>
> If your concerns have been addressed, Could you please help raise the score. If you have any other concerns, please let us know, and we will try our best to address them. Thanks.

---

> ### Author Response · Authors · 2025-11-25
> **Revisions of our Submission**
>
> Thank you again for your valuable feedback. Our revisions are as follows:
> ***
> **Revisions to Weakness 1 and Question1**
> \
> \
> In **Section 3.3(Aleatoric Uncertainty Learning: $L_{AU}$)**, we explained why the false label confidence threshold $\varepsilon=0.6$.
>
> ***
> **Revisions to Weakness 2**
> \
> \
> In **Section 5 (Conclusion)**, we mentioned that we will evaluate the performance in the case of modality loss in the future.
>
> ***
> **Revisions to Weakness 3**
> \
> \
> For **Figure 3**, we have enlarged the font size of the horizontal and vertical axes.
>
> ***
> **Revisions to Question2**
> \
> \
> In **Section 1 (Introduction)**, we explained the problem of independent estimation.
>
> ***
> **Revisions to Question3**
> \
> \
> In **Section 2 (Three Types of Uncertainty in FSTMC)**, we explained the correlation between the three types of uncertainty.
>
> ***
> **Revisions to Question4**
> \
> \
> In **Section 5 (Conclusion)**, we discussed how to extend this to classification tasks.

---

### Note · Authors · 2025-12-24

I have read and agree with the venue's withdrawal policy on behalf of myself and my co-authors.